# From Genes to Geography, from Cells to Community, from Biomolecules to Behaviors: The Importance of Social Determinants of Health

**DOI:** 10.3390/biom12101449

**Published:** 2022-10-09

**Authors:** Jaysón Davidson, Rohit Vashisht, Atul J. Butte

**Affiliations:** 1Pharmaceutical Science and Pharmacogenomics Graduate Program, University of California San Francisco, San Francisco, CA 94143, USA; 2Bakar Computational Health Sciences Institute, University of California San Francisco, San Francisco, CA 94143, USA

**Keywords:** social determinants of health, electronic health records, real-world evidence, census tract, data science

## Abstract

Much scientific work over the past few decades has linked health outcomes and disease risk to genomics, to derive a better understanding of disease mechanisms at the genetic and molecular level. However, genomics alone does not quite capture the full picture of one’s overall health. Modern computational biomedical research is moving in the direction of including social/environmental factors that ultimately affect quality of life and health outcomes at both the population and individual level. The future of studying disease now lies at the hands of the social determinants of health (SDOH) to answer pressing clinical questions and address healthcare disparities across population groups through its integration into electronic health records (EHRs). In this perspective article, we argue that the SDOH are the future of disease risk and health outcomes studies due to their vast coverage of a patient’s overall health. SDOH data availability in EHRs has improved tremendously over the years with EHR toolkits, diagnosis codes, wearable devices, and census tract information to study disease risk. We discuss the availability of SDOH data, challenges in SDOH implementation, its future in real-world evidence studies, and the next steps to report study outcomes in an equitable and actionable way.

## 1. Introduction

Understanding disease at the molecular level has dominated the field of genetics, which has been the major basis for studying disease risk over the past two decades. Researchers have presented considerable evidence that disease risk is generally conferred through genetic inheritance, and now more recently, through specific rare and common mutations [1,2]. Using tools in molecular and cellular biology, researchers and medical providers can investigate many diseases and conditions. However, the results of previous investigations have shown that disease risk is too complex to model using genetics or molecules alone. Indeed, genetic, social, and environmental factors including socioeconomic status, geolocation, and age, as well as racial and ethnic background play a role in disease risk across different population groups [3]. Growing evidence increasingly indicates the importance of accounting for the social and environmental factors that are likely to affect health outcomes. While Dr. Phil Bourne, whom this Special Issue honors, is certainly known for his work in computational methodologies and structural biology, he also understood the importance of external influences on health and called for better methods to measure and “describe individuals’ activity spaces and exposure to the built, natural, social, and economic environments that influence behaviors and health outcomes” [4].

## 2. Social Determinants of Health

Social determinants of health (SDOH) are one of the ways to capture, represent, and assess the impact of social and environmental factors in clinical research, thus improving patient care. SDOH are the conditions in which people are born, live, work, play, worship, and age, which affect a wide range of health, functioning, quality of life outcomes, and risks [5]. A patient’s SDOH can be used to estimate their access to healthcare and treatments, their positive or negative health outcomes, and to assess comorbidities by using information related to an individual’s health including alcohol and tobacco usage, socioeconomic status, insurance status, living situation, access to healthy foods, access to health literacy, and access to quality of care [5]. The main components of the SDOH commonly gathered in medicine are grouped into five domains: economic stability, education access and quality, healthcare access and equity, social and community context, and neighborhood and built environment [5]. Though noted separately, each domain is interconnected to match the complexity of SDOH variables and represent SDOH at both the population and individual levels [6,7].

Population-level SDOH measures are heavily reliant on census tract information derived from the United States Census Bureau (U.S. Census). Census tracts are indicative of geographical areas, which are defined as small, relatively permanent statistical subdivisions of a county providing information on demographic and housing estimates, occupation codes, industry codes, product and service codes, and material/fuel codes [8]. Census tracts have surveys such as the American community survey, decennial census, economic surveys, population estimates, public sector census, and economic censuses that can be leveraged to assess the overall impact of socioeconomic parameters on the health and wellbeing of patients in a given healthcare system at a given geographical location. Census tract information is gathered by assigning each person, household, housing unit, institution, farm, business establishment, or other responding entity to a specific location, and then assigning that location to a zip code tabulation area appropriate to the census or sample survey by way of geocoding [8,9]. The geocoding process ensures that the Census Bureau can provide correct counts for small geographic entities and that both the Census Bureau and data users can accumulate the data for small entities to provide totals for larger geographic entities such as zip code areas. Census tract information has been used to develop indices that directly explain the SDOH of people by using their zip code location to develop the area deprivation index, social vulnerability index, and modified retail food index [10,11,12,13]. Indices that use census tract information often categorize data by socioeconomic status, location, and education to calculate the deprivation or vulnerability of people residing in a location.

SDOH are utilized in clinical care and research studies by way of electronic health records (EHRs) which are the primary way to capture real-world data from providers on patient encounters in a health system [14]. EHRs provide a unique opportunity to study the relationship between SDOH and the management and outcomes of clinical diseases through real-world data (RWD). RWD captured in EHRs are used to develop real-world evidence (RWE) studies that analyze data and inform providers about the causes of different treatment strategies, disease risk, quality of life, and outcomes for different patients and populations. RWE studies often contain diverse patient populations that are representative of real patients’ health where common SDOH are collected. Prior to EHRs, the SDOH were primarily captured by population-level questionnaires administered by the U.S. Census or through direct questionnaires administered in clinical trials. However, the innovation of EHRs has provided us with patient-derived data to help us understand the social and lifestyle factors of patients. SDOH data coupled with questionnaires and clinical data in EHRs could be used to enable precision medical studies on healthcare access and health outcomes, by linking with data about treatments, disease conditions, drug response, insurance status, and demographics. 

Although the classification of SDOH at the individual or patient level is becoming increasingly standardized for operational and clinical research purposes, a current challenge in the wide adoption of SDOH in RWE studies is that of missing data, HIPAA regulations, and quality control issues that severely limit the amount of data available to answer clinical questions with high precision [15,16,17,18]. Therefore, the roles that the SDOH play in various chronic illnesses and diseases are ill-defined but have the potential to address population- and person-specific questions in the future. Research shows that public health goals cannot be realized without addressing the underlying SDOH that contribute to disparities and outcomes [19,20]. Therefore, healthcare research should strive to include SDOH in addition to race/ethnicity in RWE studies. A plethora of research reveals numerous socioeconomic parameters potentially accelerating disease risk, especially among minorities [20]. We must improve our understanding of the impact of SDOH on disease risk by investigating the different roles that SDOH play for patients, population groups, healthcare providers, healthcare access, and health outcomes (Figure 1).

## 3. SDOH Integration into Electronic Health Records

In EHR databases across the country, there is incompleteness of SDOH data, which has led previous RWE studies to use only race/ethnicity, sex, and age as measures of SDOH. In theory, those demographics can provide context, but cannot capture the full picture of one’s overall health. In an attempt to capture SDOH effectively in EHRs, efforts to map de-identified patients’ information to census tracts have been extremely important in providing researchers the ability to use evidence-based SDOH to answer clinical questions. However, the SDOH are often only captured in clinical notes, without structured coding, and we need better methods to obtain SDOH data trapped in notes. Currently, social aspects found in clinical notes vary across EHR databases in the country; however, the Institute of Medicine (IOM) has worked aggressively on identifying SDOH domains to be suggested for use in EHRs for academic research purposes [21]. The type of data suggested includes sociodemographic domains, psychological domains, behavioral domains, and individual-level social relationship and living condition domains. Our field is still in the early days of extracting specific SDOH information and mapping such data to and from EHRs, but alongside social indices, other structured data elements, such as insurance status, can now be used to understand a patient’s socioeconomic status. The indices can be used to answer questions related to SDOH and in cooperation with EHRs to understand surgical outcomes, drug distribution, health outcomes, and hospital readmissions. As time progresses, we will effectively utilize more SDOH data in EHRs and RWE studies.

EHR toolkits offer precise categorization of SDOH captured from census tracts for use in RWE studies. SDOH-standardized vocabularies are offered through these toolkits to map data to census tracts in EHR databases, but these are not yet widely adopted. A popular toolkit called the PhenX toolkit offers ontologies such as health insurance coverage, food insecurity, air quality index, wealth, job insecurity, food swamp, and more to increase the SDOH measures used in studies [22]. The SDOH ontologies were precisely chosen to enable highly qualitative measures that will increase the statistical power of studies [22]. The International Classification of Diseases (ICD) coding system has incorporated specific ICD-Z codes for SDOH that refer to problems related to education and literacy, housing, economic circumstances, social environments, upbringing, primary support groups, psychosocial circumstances, and occupational exposures to risk factors (Table 1). Although important, the Z codes are listed as non-diagnosis codes rather than disease-specific codes. Yet, they all play a major part in understanding the causes of health disparities in communities. At this point, there are several SDOH-standardized vocabularies and ontologies to use, but challenges occur with the utilization of these codes and capturing these codes and related data from patients in an effective way to use in research. Until EHR databases increase their SDOH data availability for SDOH-specific studies, navigating this field will remain difficult.

When SDOH are made available for research, one can start to model and predict occurrences across a wide range of diseases, relating SDOH elements to numerical measurement tests, diagnostics, and other health outcomes. It will be important to choose diseases that are easy to define in terms of severity, prevalence, and incidence of the condition, as well as diseases that have been known to be prominent in lower socioeconomic areas. With SDOH mapped to and from EHRs, we can understand what causes differences in the prevalence and incidence of a disease between different population groups. To do this, we can use descriptive biostatistical methods such as ordinal logistic regression and multivariate logistic regression that describe the relationship between categorical variables predominantly found in EHR datasets, alongside other effective analysis techniques for EHRs as previously described [24]. 

## 4. The Future of SDOH in Real-World Evidence Studies

The next step in biomedical informatics is to develop a streamlined method for mapping census tract information to EHRs that can be implemented by each institution. In structured data, there needs to be a unified goal to create more diverse databases by providing healthcare access to underserved populations, thus ensuring that SDOH studies are well-balanced and produce answers backed by realistic outcomes. In the future, we could increase the versatility of SDOH for medical studies by developing a universal “medical grade social index” that could contain specific SDOH information that is most directly related to clinical outcomes, to best answer research questions. The power of this data will be to create knowledge that fills current gaps in healthcare, such as the need to incorporate brick-and-mortar needs-based healthcare services into deprived communities.

The future looks bright for even newer sources of data on SDOH. Such patient-derived data have allowed us to obtain information on SDOH beyond that in EHRs thanks to people using health measures including wearable fitness monitors, smartphones, step trackers, food trackers, and telemedicine. Tracking information from patients allows us to understand patients’ daily activities that can lead to chronic illnesses and diseases, and thus understand how to potentially eliminate them. Currently, however, patient-derived data offer limited information across larger populations. Luckily, the adoption of mHealth has increased SDOH measures by utilizing mobile technology to improve health goals in communities. mHealth is a medical and public health practice supported by mobile devices, such as mobile phones, patient monitoring devices, personal digital assistants, and other wireless devices, as defined by the World Health Organization [25]. Whilst mHealth is not widely adopted yet, patient-derived data can be used in the future to estimate a patient’s health or risk of clinical diseases and provide better patient-centered health approaches to improve overall individual health. How mHealth and digital devices will feed into SDOH is still an active area of research. Other newer sources of SDOH-related data will also include genomic information for large populations uploaded to EHR databases, which is already being implemented by various institutions around the country [26].

Thus far, questionnaires, population surveys, and patient-reported outcomes are heavily used in SDOH and epidemiological research studies because they allow researchers to answer clinical research questions when structured data elements of the EHR database system themselves do not give a full picture of SDOH data. Due to inconsistencies in SDOH data availability, the field has been using questionnaire-based data to answer in-depth questions related to SDOH, along with EHR data, to understand surgical outcomes, drug distribution, health outcomes, and hospital readmissions [27,28,29,30].

## 5. Conclusions

The communication of RWE studies, especially related to disparities in health, has the potential to create political conflict, medical mistrust, and harm to marginalized communities. Therefore, it is important to effectively communicate results in a manner that is understood by both clinicians and patients from all paths of life so that the knowledge of health outcomes is more accessible. More importantly, including a diverse set of researchers to harness the power of this data is important for leveraging different mindsets to understand SDOH clinical questions in a different context. Since we know that a large portion of health disparities occur in communities under-represented in medicine, the best way to relate to these communities is by having people who understand the vernacular, culture, and the patience needed to communicate risk, health literacy, and health accessibility to the most affected communities.

All of this is an acknowledgment and perhaps a frustration that studying SDOH is likely to be harder than studying genes, molecules, and cells. However, the more we finetune the idea and execution of SDOH in EHR databases across the country, the more we will increase the opportunities to use precision medicine to target clinical diseases. Precision medicine can be used to accurately prescribe patients based on RWE of health outcomes and treatment patterns respective to different population groups. More importantly, it will give clinicians the ability to prescribe patients using patient-centered approaches derived from research. Nevertheless, it is critical to remember that precision medicine means more than just genes, molecules, and cells. The future implementation of SDOH will provide greater precision of treatments based on an array of demographics, lifestyle factors, and environmental factors, all of which are likely to make a greater difference for our patients than any given measured cell or base-pair in their genome.

## Figures and Tables

**Figure 1 biomolecules-12-01449-f001:**
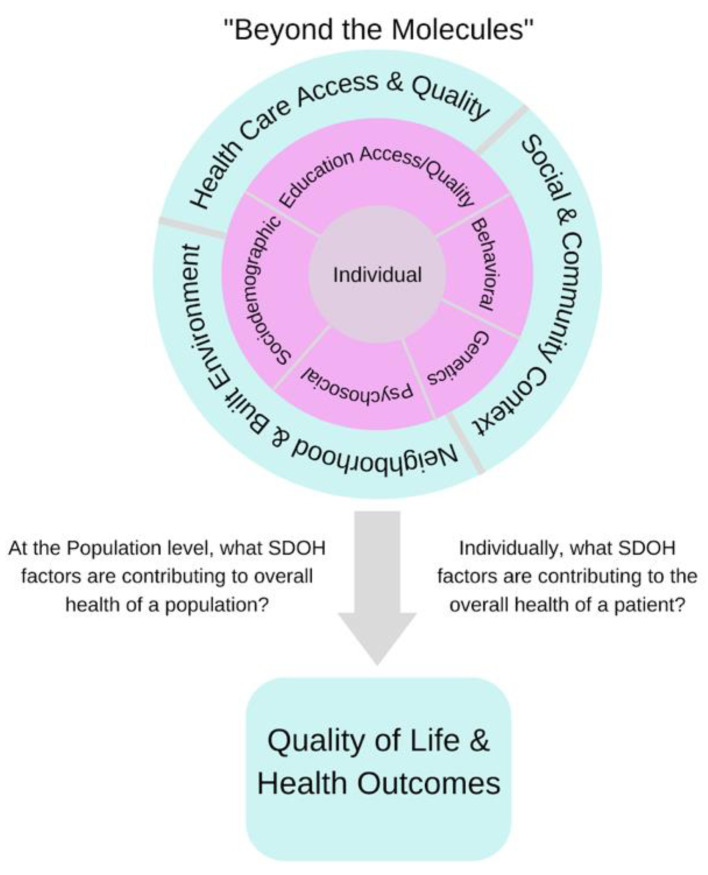
Grouped SDOH factors are categorized at the population and individual levels. At each level, we define the SDOH factors that contribute to the overall health of a population and the overall health of a patient, which mark the difference between a good outcome and a poor outcome.

**Table 1 biomolecules-12-01449-t001:** International Classification of Disease (ICD) Z codes for SDOH [23].

ICD-10 CM Code	SDoH Categories
Z55	Problems related to education and literacy
Z56	Problems related to employment and unemployment
Z57	Occupational exposure to risk factors
Z58	Problems related to physical environment
Z59	Problems related to housing and economic circumstances
Z60	Problems related to social environment
Z62	Problems related to upbringing
Z63	Other problems related to primary support group, including family circumstances
Z64	Problems related to certain psychosocial circumstances
Z65	Problems related to other psychosocial circumstances

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
