# Peer review of "From Genes to Geography, from Cells to Community, from Biomolecules to Behaviors: The Importance of Social Determinants of Health"

_biomolecules, 2022, doi:10.3390/biom12101449_

Round 1

Reviewer 1 Report

Authors describe the necessity of the "social determinants of health (SDOH)" to be collected for patients in studies. The manuscript is a state of art oriented in the need of these informations to be collected.

The paper is well written.

I have two important points:

1) The paper lacks a crucial information: Authors do not mention the statistical methods required for analysis of this important amount of data.

2) How would be reported all these informations? as PRO? by physician? This is a crucial information as for PRO, it is well known that if some patient do not report, this is not randomly.

Typo: There is an apostrophe in the email of the corresponding author.

Author Response

Authors describe the necessity of the "social determinants of health (SDOH)" to be collected for patients in studies. The manuscript is a state of art oriented in the need of these informations to be collected. Thanks to Reviewer 1 for your comments. In the updated revision we address the statistical methods that can be used for EHR data analysis, and how the research can be reported.
The paper lacks a crucial information: Authors do not mention the statistical methods required for analysis of this important amount of data. To make this more clear, I have added text on page 4 line 161-66. Newly added text explains methods predominantly used to identify statistical significance amongst categorical variables predominantly found in EHRs.
How would be reported all these informations? as PRO? by physician? This is a crucial information as for PRO, it is well known that if some patient do not report, this is not randomly. In the text on page 5, line 195-98 we state that currently SDOH information is reported through questionnaires, population surveys, and patient reported outcomes. We have now made this text more clear.

Reviewer 2 Report

This paper addresses the inclusion of social/environmental factors and health outcome for populations and individuals in genomics research by integration of social determinants of health in electronic health records.
The research design is appropriate. The classification of social determinants of health and related issues is properly considered and methods for its integration into electronic health record by specific tool and toolkits are considered. Available questionnaires population survey and patients reported outcomes are singled out. The paper does not present results in a strict sense but provides useful suggestions for work that would be very worthwhile for the availability and implementation of social determinants of health data.
The paper provides a good basis for future development.

Author Response

This paper addresses the inclusion of social/environmental factors and health outcomes for populations and individuals in genomics research by integration of social determinants of health in electronic health records.

Thanks to Reviewer 2 for your comments!

The research design is appropriate. The classification of social determinants of health and related issues is properly considered and methods for its integration into electronic health record by specific tool and toolkits are considered. Available questionnaires, population surveys and patients reported outcomes are singled out. The paper does not present results in a strict sense but provides useful suggestions for work that would be very worthwhile for the availability and implementation of social determinants of health data.

The paper provides a good basis for future development. 

Thank you for the positive comment!

Round 2

Reviewer 1 Report

The authors have taken into account my comments.

The typo on the corresponding author mail is still present.

"jayso'[email protected]" --> [email protected] ?